# Monitoring of Air-Dispersed Formaldehyde and Carbonyl Compounds as Vapors and Adsorbed on Particulate Matter by Denuder-Filter Sampling and Gas Chromatographic Analysis

**DOI:** 10.3390/ijerph16111969

**Published:** 2019-06-03

**Authors:** Stefano Dugheri, Nicola Mucci, Giovanni Cappelli, Alessandro Bonari, Giacomo Garzaro, Giorgio Marrubini, Gianluca Bartolucci, Marcello Campagna, Giulio Arcangeli

**Affiliations:** 1Industrial Hygiene and Toxicology Laboratory, Careggi University Hospital, 50134 Florence, Italy; 2Department of Experimental and Clinical Medicine, University of Florence, 50134 Florence, Italy; nicola.mucci@unifi.it (N.M.); giovanni.cappelli@unifi.it (G.C.); giulio.arcangeli@unifi.it (G.A.); 3General Laboratory, Careggi University Hospital, 50134 Florence, Italy; bonaria@aou-careggi.toscana.it; 4Department of Public Health Sciences and Pediatrics, University of Turin, 10126 Turin, Italy; giacomo.garzaro@unito.it; 5Department of Drug Sciences, University of Pavia, 27100 Pavia, Italy; giorgio.marrubini@unipv.it; 6Department of Neurosciences, Psychology, Drug Research and Child Health Section of Pharmaceutical and Nutraceutical Sciences, University of Florence, 50134 Florence, Italy; gianluca.bartolucci@unifi.it; 7Department of Medical Sciences and Public Health, University of Cagliari, 09124 Cagliari, Italy; mcampagna@unica.it

**Keywords:** formaldehyde, carbonyl compounds, aldehydes, air pollution analysis, environmental analysis, 2,4-dinitrophenylhydrazine derivatization, PM_2.5_

## Abstract

Carbonyl compounds (CCs) are products present both as vapors and as condensed species adsorbed on the carbonaceous particle matter dispersed in the air of urban areas, due to vehicular traffic and human activities. Chronic exposure to CCs is a potential health risk given the toxicity of these chemicals. The present study reports on the measurement of the concentrations of 14 CCs in air as vapors and 2.5 µm fraction PM by the ENVINT GAS08/16 gas/aerosol sampler, a serial sampler that uses annular denuder, as sampling device. The 14 CCs were derivatized during sampling prior to gas-chromatographic separation and multiple detection by mass spectrometry, nitrogen-phosphorus thermionic, electron capture detection. Outdoor air multiple samples were collected in four locations in the urban area of Florence. The results evidenced that formaldehyde, acetaldehyde, and acetone were the more abundant CCs in the studied areas. The data collected was discussed considering the particle to vapor ratio of each CC found. The CCs pollution picture obtained was tentatively related to the nature and intensity of the traffic transiting by the sampling sites. This approach allowed to determine 14 CCs in both concentrated and diluted samples and is proposed as a tool for investigating outdoor and indoor pollution.

## 1. Introduction

Air pollution has been associated with a variety of adverse health effects [1] and is believed to be one of the major causes of premature deaths worldwide [2]. Carbonyl compounds (CCs) are ubiquitous components of the environment and are present both as gases and as condensed species adsorbed on the surface of particle matter (PM) dispersed in the air. Human exposure represents a potential health risk, given the well-documented toxicity of these chemicals. Despite a significant health risk due to exposure, the mechanisms of CCs toxicity are poorly understood. LoPachin et al. [3]—based on the hard and soft, acids and bases theory [4]—indicate that short-chain and longer-chain saturated aldehydes are hard electrophiles that become toxic by forming adducts with hard nucleophiles present in biological systems. In contrast, α,β-unsaturated carbonyl compounds and α-oxoaldehydes are soft electrophiles that preferentially react with soft nucleophilic thiolate groups of cysteine residues. Several authors [5,6,7] propose that environmentally derived aldehydes can accelerate diseases by interacting with endogenous aldehydes generated during oxidative stress. Formaldehyde (FA) is the most important carcinogen in outdoor air among the 187 hazardous air pollutants (HAPs) identified by the United States (US) Environmental Protection Agency (EPA). It is by far the most important HAP in terms of health risk, accounting for over 50% of the total HAPs-related cancer risks in the US [8].

Aldehydes and ketones are significant constituents of atmospheric carbonaceous PM formed as a result of primary emissions and secondary atmospheric reactions. Due to the high Henry’s law constant, carbonyls may dissolve in the liquid water content of atmospheric particles. Water-soluble fraction is a part of PM which contains inorganic ions and water-soluble organic compounds, that due to hydrophilicity, are suggested to serve as a newly formed cloud condensation nuclei [9]. This fraction is more easily absorbed by the body and therefore, the toxicity of PM is increased [10,11,12]. Particles emitted from combustion processes, where diesel engines are major contributors, are among the most common emissions in populated areas [13]. In urban areas, motor vehicle exhaust became an important source of aldehydes in air both through direct emission of aldehydes and through the emission of hydrocarbons, which in turn were converted to aldehydes through photochemical oxidation reactions [14]. The most recent Urban Air Toxics Study results posted on the EPA web site include FA, acetaldehyde (AA), and acrolein (ACR) as significant contributors to the summed risk values for mobile sources of air toxins. Diesel exhaust particles as well as the gas-phase of diesel exhaust fumes are believed to play a key role in carcinogenic effect on human health [15]. The International Agency for Research on Cancer classified diesel exhaust as carcinogens to humans. Moskal et al. [16] reported that approximately 20% of the amount of aerosol inhaled through the nose penetrate the respiratory system when the aerosol is made by fractal-like aggregates having dimensions in the range from 1.7–2.1 µm and radius of gyration of 0.24–0.36 µm. A weak dependence of deposition efficiency on fractal dimensions and the radius of gyration values of the aerosol particles was also documented. Alföldy et al. [17] studied diesel exhaust particles emitted by Euro 1, 2, 4 light-duty diesel vehicles to indicate respiratory tracts deposition fraction. With the increasing use of alternate and reformulated automotive fuels, an additional source of aldehyde emission is now in evidence. Depending on the type of oxygenated compounds added to the automotive fuels, an increased number of CCs are now emitted in automobile exhaust fumes [18]. Grosjean et al. attributed the change in ambient AA/FA ratio to a change in reliance on ethanol as vehicle fuel in Brazil [19].

The currently lowest guideline environmental values for CCs set at 3 µg/m^3^ was published in 2016 by the California Air Resources Board as the Chronic References Exposure Limits [20]. For FA the European Commission established in November 2014 an outdoor air limit of 1 µg/m^3^ [21]. Hourly measurements are required for air toxics assessment by the 1990 US Clean Air Act Amendment [22]. To date, only a few analytical methods can reach these very restrictive limits, and there is thus a need for comprehensive methods able to analyze airborne FA, as well as other carbonyl pollutants in living environments. The standard methods use a chemical reaction between carbonyls and 2,4-dinitrophenylhydrazine (2,4-DNPH) to produce strong UV-absorbing derivatives for sampling atmospheric carbonyls prior to their separation and determination by liquid chromatography (LC). Moreover, aldehydes and ketones have been scarcely determined in PM. Toda et al. [23] define “a mystery” the amounts of CCs and specifically FA associated with liquid water content of PM. Conventional methods using cartridges impregnated with 2,4-DNPH are not able to differentiate the two sample fractions represented by the air-dispersed CCs in the vapor phase and those adsorbed on particulate matter. Thanks to the use of the denuder-filter sampling technique instead, it is possible to collect the two fractions separately [24,25].

In other studies, PM samples for CC analysis were collected directly on filter and derivatized on-solvent after sampling [12,26]. Chromatographic separation of CCs, with prior derivatization to form the 2,4-dinitrophenylhydrazones derivatives, has been used for the last twenty years, and nowadays, it is applied in air monitoring programs [12,27,28,29,30,31]. Previous authors found, however, that the 2,4-DNPH LC method for aldehydes does not comply with the 30% minimum performance criteria at the sub µg/m^3^ level [29,32,33]. In order to investigate the occurrence of carbonyl groups present in PM, various analytical protocols have been used including LC [12,26] and capillary chromatography [34,35,36], as well as direct analysis and multi-step derivatization prior to gas chromatography (GC)/mass spectrometry (MS) analysis [9].

This study is the first to report the determination of 14 CCs in gaseous and 2.5 µm fraction PM (PM_2.5_) by applying the ENVINT GAS08/16 Gas/Aerosol sampler, a new and selective denuder-filter sampler. The PM_2.5_ was chemically desorbed by an innovative xyz on-line GC autosampler. So, the large volume of extraction solvent required could be injected almost entirely in a GC by Large Volume Injector (LVI) techniques. In addition to the use of MS and nitrogen-phosphorus thermionic specific detector (TSD), we introduced an electron capture detector (^63^Ni ECD) in the system to improve the sensitivity of detection. We demonstrated that the 2,4-dinitrophenylhydrazones, are amenable to GC and possess strong electron-capturing properties. Finally, the study reports the results of a campaign of environmental monitoring conducted in the streets of Florence - recording the number and types of vehicles - to enable the hourly determination of ambient gaseous and PM_2.5_-adsorbed CCs.

## 2. Materials and Methods 

### 2.1. Sampling

#### 2.1.1. Sampling Equipment

The ENVINT GAS08/16 gas/aerosol sampler (Envint, Montopoli di Sabina, Italy) consists of a housing that accommodates eight denuder lines (Figure 1). 

The lines are accommodated in sampling positions selected according to the time sequence specified in the programming operations. Denuders collect reactive gases, while a series of filters capture PM components. After sampling, each line is removed from the housing for analysis. Before and after sampling the denuders are protected from passive contamination by external air by a Teflon ball valve which opens when the sampling starts. Programming of the sampler is carried out by means of a tablet working in Android OS. The user can set the sequence of sampling, time of sampling, and flow rate through a Wi-Fi connection. The system records the sequence of sampling, start- and end-time of each sampled channel, ambient temperature (T), pressure (P) and relative humidity (RH) of external air, flow rate (at standard T and P), pressure drop on filter pack, the volume of sampled air. All information is stored in the internal memory of the sampler and transmitted via Wi-Fi to users during a visit at the site. The system is programmed in order to continuously check the electronic and pneumatic part of the sampler.

The annular denuder is made of concentric cylinders of size 150 x 16.2 mm (length x diameter, annular space: 16,2–15 mm) made of glass specifically developed for the collection of gas. When air flows through the annular denuder coated with 2,4-DNPH, the gas molecules diffuse on the walls and are removed from the airstream. PM_2.5_ proceeds unaffected and is collected on the glass fiber filter pack impregnated with 2,4-DNPH.

The sampling pump is a sealed membrane pump set to a constant flow rate of 1.0 L/min.

#### 2.1.2. Coating of the Annular Denuder

A 2,4-DNPH solution (Cat. No. D199303, Aldrich, Sigma-Aldrich, St. Louis, MO, USA) is used to coat the denuder’s walls. The walls of the denuder are etched to increase the surface area and obtain an efficient coating with a solution of recrystallized 1% 2,4-DNPH and 1.5% orthophosphoric acid in acetonitrile (ACN). The coating was provided by pouring l mL of this solution into the denuder, which was slowly rotated and inclined to ensure complete wetting of the walls. The solution was then evaporated using a clean airstream. During this operation, the denuder was continuously rotated in order to yield a homogeneous film of 2,4-DNPH. Field samples were collected with a train of two denuders. One additional annular denuder coated with potassium iodide (10 mL of 10% KI solution) was placed in front of the 2,4-DNPH coated ones, to remove interfering ozone [37,38,39]. When dried, the denuders were sealed with threaded Teflon caps until use.

#### 2.1.3. Preparation of the Glass Fiber Filters

2,4-DNPH was recrystallized twice from a mixture of water/ACN (5:1, v/v) before use. The 2,4-DNPH coating solution (500 ppm) used for air sampling was prepared by mixing 0.1 g purified 2,4-DNPH and 1 mL orthophosphoric acid in 200 mL ACN. Glass fiber filters (37 mm) were soaked in the 2,4-DNPH coating solution for 30 min and then dried overnight under vacuum.

#### 2.1.4. Gravimetric Determinations

The collected aerosol mass was determined by weighing all filters before and after exposure using a Sartorius Cubis Ultra Micro Balance (Cat. No. MSA2.7S-000-DM, Sartorius AG, Göttingen, Germany) in the temperature- and humidity-controlled SCC 400L climate cabinet (Sartorius AG, Göttingen, Germany) at 20 ± 1 °C and 50 ± 2% relative humidity. All weighing was made before and after conditioning filters for three days in a save-filters holder used also for transport.

### 2.2. Sampler Desorption and Analysis

#### 2.2.1. Online xyz Autosampler

A new fully automated Flex xyz autosampler (EST Analytical, Fairfield, CT, USA) was coupled with a PTV-GC/Saturn 2200 MS, TSD and ^63^Ni ECD (Varian Analytical Instruments, Palo Alto, CA, USA) for analysis of vapor and PM_2.5_. The robotic sampling platform provides fully automated functions including the desorption of the denuders and glass fiber filters, the removal of the excess of 2,4-DNPH reagent and the injection into the GC apparatus using the Multi Tool eXchange system (MTX). The MTX fills a 5.0 mL syringe equipped with a PTFE plunger and a needle tip (Cat. No. 2600040, ILS Innovative Labor Systeme GmbH, Stützerbach, Germany) with 5.0 and 3.0 mL of ethyl acetate required for carbonyl-2,4-dinitrophenylhydrazones desorption from the denuder and 37-mm glass fiber filter, respectively. The eluate of the denuder was collected in 10 mL clear headspace screw cap vials (Cat. No. VX0018-2346CTC, CPS Analitica, Milano, Italy). The filters were put into 8-mL vials (Cat. No. 090.004.0053, Soltec, Milan, Italy), equipped with a calibrated filter positioner (Cat. No. 090.004.0054, Soltec, Milan, Italy), and sonicated by Sonica extractor (Soltec, Milan, Italy), on-line with Flex autosampler. The eluted solutions were thus viaapplied onto an Oasis MCX Plus cation exchange cartridge (Cat. No. 186003516, Waters, Milford, CT, USA) to remove the excess of derivatization agent, and collected in clear headspace screw cap 20-mL vials (Cat. No. VXX018-2375CTC, CPS Analitica, Milano, Italy) and 4-mL amber vials (Cat. No. 27006, Supelco, Bellefonte, PA, USA). Ten microliters of internal standard (IS) solution containing 20 ng/mL of diphenylamine (Cat. No. 24,258-6, Aldrich, Sigma-Aldrich, Saint Louis, MO, USA) and 10 ng/mL of 2-chlorobenzaldehyde (Cat. No. 12,497-4, Aldrich, Sigma-Aldrich, Saint Louis, MO, USA) were added. A one-mL syringe (Cat. No. 8131, SGE Analytical Science, Trajan, Ringwood, Australia) was used for LVI injection into the GC system.

#### 2.2.2. PTV-GC/MS/63Ni ECD/TSD Instrument

The calibration standards and eluates were injected into a Varian CP-3800 GC with two 1078/1079 capillary injectors (Scion Instruments, Livingston, UK) equipped with CO_2_ coolant, 3.4 mm, 5.0 mm, and 54 mm Siltek deactivated glass frit liners (Cat. no. 21709-214.5, Restek Corporation, Bellefonte, PA, USA) and two Merlin Microseal septum-less systems (Merlin Instrument Co, Newark, NJ, USA).

To inject large amounts of ethyl acetate containing the analytes into the 1078/1079 PTV inlet, the solvent must be eliminated instantly while retaining all the target analytes in the liner. To achieve this effect during injection, the split vent valve must remain open while a large stream of carrier gas removes the solvent. After complete removal of the solvent, the split vent valve is closed, and the inlet temperature increased. The analytes inside the liner are then rapidly vaporized and transferred into the column. The split vent is then opened again to remove the remaining traces of solvent vapor from the liner. For the injection of volumes >3 μL, fine-tuning of the PTV conditions is necessary to minimize the loss of analytes without sacrificing the chromatographic performance. Sixty µL was chosen as the optimal volume for at-once injection, combined with a split rate of 250:1, with 5 µL/s injection for a solvent-venting of 2.00 min.

A temperature ramp mode for the two PTV injectors was performed to vent the solvent vapor and inject the analytes into two 30 m, 0.25 mm, 0.25 µm DB-35MS UI columns (35% phenyl, 65% methylpolysiloxane, Cat. No. 122-3832UI, Agilent J&W GC Column, Santa Clara, CA, USA). The first chromatographic column was connected to the MS detector while the second was connected to ^63^Ni ECD and TSD with 1 meter, 0.25 mm intermediate polarity fused silica (Cat. No. 25727, Supelco, Sigma-Aldrich, Bellefonte, PA, USA) by a fused silica Y-connector (Cat. No. 23632, Supelco, Sigma-Aldrich, Bellefonte, PA, USA). The column temperature was initially held at 40 °C for 1.5 minutes, increased to 250 °C at a rate of 10 °C/min and then held for 10 minutes. Helium was used as the carrier gas, at a flow rate of 1.2 mL/min for MS, and at 2.0 mL/min for ^63^Ni ECD and TSD detection. 

### 2.3. Vapor Generation 

For the generation of samples containing known concentrations of CCs that could model convincingly actual air samples, we made use of the static system proposed by Pieraccini et al. with modifications [40]. A volume of 5µL of aqueous solution of known CCs concentration (0.4–51.2 μg/μL) was injected using a 10μL GC syringe into the injector port at 200 °C of a modified Adsorbent Tube Injector System (ATIS, Cat. No. 28521, Supelco, Bellefonte, PA, USA) and collected into 100 L Tedlar sampling bags (Cat. No. KB3-50, Sensydine, St. Petersburg, FL, USA). The CCs air concentration (CCsCair) was calculated according to the following formula:CCsCair = Csol/V(1)
where Csol is the mass of the CCs in the aqueous solution injected (μg), and V is volume (L) of the air. The concentration of water vapor produced by the impinger was determined by measuring the dew point temperature with a photoacoustic infrared Innova type 1312 Multigas Monitor (LumaSense Technologies, Santa Clara, CA, USA). Atmospheric pressure was determined with a GE Druck DPI 705 digital pressure indicator (General Electric, Boston, MA, USA).

Calibration jar (Sensidyne, part. No. 7013376, St. Petersburg, FL, USA) allows proper placement of the ENVINT GAS08/16 Gas/Aerosol sampler between the sampling pump (1.0 L/min) and the 100 L Tedlar sampling bags. 

To calculate how long it takes for the Tedlar sampling bag vapor to fill the calibration jar, the following equations were used: C = C_0_ (1 − exp(Q·t/V))(2)
t = 2.303 V/Q log (C_0_/C_0_ − C)(3)
where, t is the time (min) to achieve the concentration C from 100 L Tedlar sampling bag to the calibration jar, C_0_ is the original concentration into the calibration jar, V is the jar volume (L), Q is the flow rate (L/min) of the sampling pump, and 2.303 is the divisor to convert the natural logarithm to the base 10 logarithm after algebraic processing of Equation (2).

### 2.4. Particles Generation

The mass deposition of particles in a vertically mounted annular denuder was investigated using the dynamic system. We used an atomizer aerosol generator model 3079A (produced in Germany by TOPAS GmbH, Dresden, Germany and marketed by TSI Incorporated, Shoreview, MN, USA) with a stainless-steel twin-stream injection nozzle to produce PM_2.5_. The particles were collected on a back-up filter placed at the denuder 3 outlet. 

### 2.5. Quantitative Analysis

Five-point calibration curves were constructed by plotting the ratio of the base peak area of the aldehyde/ketone DNPH mix (Cat. No. CRM4M7285, Sigma-Aldrich, Saint Louis, MO, USA) to the base peak area of the corresponding ISs. Diphenylamine was used for MS and TSD while 2-chlorobenzaldehyde was selected for ^63^Ni ECD. A linear regression plot was generated, and the instrumental limit of detection (LOD) was computed as LOD = 3SE (y-axis intercept)/slope, where SE is the standard error of the y-axis intercept, and slope is the mean slope of the calibration curve. The limit of quantification (LOQ) was estimated in the same way using 10 as a coverage factor, which corresponds to 3.3-times the LOD.

Quantitation was thus obtained measuring the ratios of the corresponding peak areas for each target analyte in the chromatograms of the samples and computing the analytes levels through the calibration curve equations.

### 2.6. Sampling Sites

Sample collection can be divided into three different categories: background samples, vehicular emission (VE)-dominated, and roadside. As VE-dominated sampling sites, we selected large parking areas around the Careggi hospital, where the number of cars in transit and parked is considerable. The roadside sample sites were roads characterized by moderate traffic, running near the parking lots.

There were many pre-Euro diesel-fueled buses and heavy-duty vehicles on the road in Florence. According to the Autoritratto 2017, released by Automobile Club Italia, the percentage of motorized vehicles in Florence’s municipality was as follows: out of 297,467 vehicles registered, 50% were liquefied petroleum gas (LPG) fueled and 50% were gasoline-fueled. Of these vehicles, including cars, heavy and light industrial vehicles, motorcycles and buses, 163,907 are category Euro 4 and higher (i.e., engines equipped with a particulate filter exhaust), while the remaining 133,560 vehicles are of category 3 Euro and lower. A total of 12 VE-dominated, eight roadsides, and six background samples were collected during the winter of 2018. The monitored roads are those that surround the Careggi University Hospital. In particular Via Giulio Caccini, Via delle Gore, Via Gaetano Pieraccini and Via delle Oblate, in order to assess the influence of CCs ‘outdoor pollution on indoor environments (Figure 2). 

The samples for each VE-dominated source were collected when a maximum number of vehicles transit the sites. For the roadside samples, the collections were carried out during rush hours (slow traffic), but no traffic jam was observed. This avoids different carbonyls emission ratios from the idle engines. The background samples were collected in the daytime, consistently with the sampling periods for both VE-dominated and roadside samples. The samplers were fixed at ground level with an inlet at a height of 1.5 m (breathing zone).

## 3. Results and Discussion

The aim of our work was to evaluate an innovative monitoring approach for measuring atmospheric CCs in dust and vapor that is simple, fast, and sensitive. To come up with a successful protocol, two fundamental requisites had to be met. First, we tested a new denuder-filter sampler which is able to sample gases and particles. Second, we used chemisorption with 2,4-DNPH, the most commonly used derivatization agent for CCs used in industrial hygiene and environmental monitoring. Derivatization, however, necessarily involves several steps in the investigation, including the preliminary estimate of the amount of the CC gaseous component that will be collected into the denuder under various sampling conditions, the reagent’s blank value, and the removal of the 2,4-DNPH unreacted reagent.

The use of GC had an important role in the present study for monitoring emissions into the atmosphere. We proposed here a development in the injection technology, a capillary column that widens the range of separation, and an improvement in detection sensitivity afforded by the selection of multiple detectors coupled together.

### 3.1. Performance of Denuder-Filter Sampler 

Denuders were first introduced by Crider et al. [41] in the late 1960s to determine sulfate in aqueous solution. In 1983 annular denuders were applied for accurate measurements of the gas-particle distribution of volatile compounds [42]. Since 1983, annular denuders have been widely used for sampling ambient air pollutants. Unlike silica gel and C18 silica gel cartridges impregnated with acidic 2,4-DNPH, annular denuders coated with 2,4-DNPH have been rarely used to collect CCs.

Collection efficiencies (E) of the 2,4-DNPH-coated annular denuders for CCs were calculated as
E = 1 − m_2_/m_1_(4)
where m_2_/m_l_ is the ratio of the peak areas of the same hydrazone derivative extracted from the second and the first denuder, respectively.

Each result represents the average value of five replicates collected at a given air-flow rate. To compare the experimental with theoretical data it is important to remember that in the case of perfect sorption the fractional penetration of a gaseous species with diffusivity D passing through an annular denuder is described by the Gormley and Kennedy equation [24,42,43,44,45]
m_2_/m_1_ ≈ 0.82 exp (−22.53Δ)(5)
with Δ is defined as
Δ = (πDL/4F) (d_1_ + d_2_)/(d_2_ − d_1_)(6)

In Equation (6), L is the denuder length, F the air flow rate, d_l_ is the internal diameter of the outer cylinder of the annular section, and d_2_ is the external diameter of the inner cylinder of the annular section, respectively. 

Hence Δ depends only upon the parameters L and F. The choice of the value of d is subject to the following conditions, which ensure a laminar flow field [46]: Reynolds number, Re ≤ 2000 and the length l of the uncoated part of the tube which is needed to establish a laminar air flow l ≥ 0.05d Re.

Thus, in a laminar flow through an annulus, Δ may be optimized by varying not only the tube length and the air flow rate but also the width of the annular section and the diameter of the inner tube. By reducing the equivalent diameter of the annulus and increasing its internal diameter, values of Δ much larger than those corresponding to Δ at given ratios F/L are obtained. Moreover, a short annular denuder may give high Δ values even at high flow rates without any alteration of the laminar regime, being Re-dependent on the internal diameter of the annulus. For instance, if an annular denuder with d_1_ = l.0 cm and d_2_ − d_1_ = 0.3 cm is used, a laminar flow up to 33 L/min with Re equal to 2000 can pass through it. For a cylindrical tube with the same equivalent diameter, the limit of laminarity is at about 4.3 L/min [42].

The interference due to the deposition of atmospheric particles on the denuder walls must also be considered. In the laminar flow regime, two major effects take place, sedimentation and Brownian diffusion. While the former is eliminated by setting the denuder in a vertical position, the latter depends upon the particle diffusion coefficients. For 0.01 to l.0 µm diameter aerosols, these coefficients range from 5.2·10^−4^ to 2.7·l0^−7^ cm^2^/s. These values are much lower than those of gaseous species. Since the equation describing the fractional penetration of a monodisperse aerosol through an annular denuder has the same form as that of a gas, we can predict that particle deposition should be negligible during sampling of aerosol at concentrations environmentally relevant.

Actually, from the denuder’s vapor collection efficiency study, we observed that experimental data obtained by loading a standard CCs vapor solution through two denuders put in series (m_1_ and m_2_), and then detecting the separate concentration of the CCs in m_1_ and m_2_ agree well with those calculated from Equation (4) for FA. Collection efficiencies for the other C_2_-C_3_ CCs were lower but still at satisfactory levels (Table 1). In addition, from the results obtained by particles generation from five replicate measurements, we observed that the average mass of deposited particulate on the denuder was negligible (<0.3%) with respect to the mass collected on the glass fiber filter.

Annular denuders were found to be superior in efficiency when compared with other sampling devices if the same flow rates were used. The possibility of operating with complete efficiency at an air flow rate of 1 L/min and with a 90% recovery for AA and propionaldehyde (PA) even at 2 L/min, appreciably improves the sensitivity for ambient monitoring of low-boiling point aldehydes. Experimental data indicated that the increase in humidity from 50 to 90% did not affect the denuder efficiency. Low CCs blank concentration both in denuders (0.015 µg/cartridge) and in filters (0.012 µg/cartridge) allowed high sensitivity.

The results indicate that concentrations of the order of 0.4ppbv for formaldehyde and 0.2ppbv for C2-C3 aldehydes can be easily measured when 60 1itres of air are collected and, more importantly, that no changes in efficiency are observed when aldehyde content and relative humidity vary (50%–90% of humidity variation) during vapor generation experiments.

### 3.2. Selection of the Analytical System

The analysis of 2,4-DNPH derivatives of environmentally relevant CCs suffers from one major problem that is that the excess of the derivatizing agent must be removed from the sample prior to analysis. In this study, a cleanup of the sample by cation exchange solid phase extraction was adopted before GC analysis. We observed that by removing the excess of 2,4-DNPH reagent using a polymeric MCX Plus Oasis mixed-mode cation-exchange cartridge not only did the LOQ drop by one order of magnitude but also liquid chromatography (LC) and GC analyses could be coupled with single- and triple-quadrupole MS. To our knowledge, few studies focused on the quantitative analysis of carbonyls by LC–MS/MS [47,48]. In agreement with the results reported by Van den Bergh [49], we observed that both electrospray ionization (ESI) and atmospheric pressure chemical ionization (APCI) in the negative-ion mode were suitable for the detection of the 2,4-DNPH derivatives of CCs. However, the risk for LC coelutions of derivatized CCs increases with the number of undesired side-reaction products and syn/anti 2,4-DNPH isomers. Moreover, the purchase and maintenance costs for the LC system are much higher than that of the GC. Consequently, we selected the cheaper and easier-to-operate system consisting of a GC-ECD/TSD apparatus and a GC-MS, since these instruments permitted the separation of the major 2,4-DNPH degradation product (2,4-dinitroaniline) from the FA-2,4-dinitrophenylhydrazone and from all 2,4-dinitrophenylhydrazones including their isomers.

The use of GC-ECD/TSD and MS systems for the determination of the 2,4-dinitrophenylhydrazone carbonyl-derivatives is widely used and applied in the analysis of various matrices. The ECD detector allows to achieve the limit of detection lower than 1 ng/mL, about ten times lower respect to that of the TSD. In fact, the aromatic moiety of the derivatizing agent is strongly deactivated by the substitution of two nitro-groups, that increase the positive charge residue on the ring. Therefore, after removing the excess of 2,4-DNPH the background noise is greatly reduced, and the CCs derivates are easily detected by ECD detector.

On the other hand, the use of the MS detector allows high specificity of the analytical signal, with the acquisition and comparison between MS spectra. However, splitting up of the signal intensity to the all fragmented ions decreases the sensitivity of detection.

The calibration parameters (LOD, slope, intercept, and standard error) obtained by using the GC-ECD/TSD and MS systems in the determination of studied analytes document these observations and are compared in Table 2.

As can be seen by observing the data in Table 2, the calibration of the three GC techniques evidence the fact that the y-axis intercept is always of the same magnitude (~0.8 CC/IS relative detection units) and statistically not different from 0, meaning that no systematic error occurred in the calibration (compare the SE with the y-axis intercept value for each CC). As far as the sensitivity of the three techniques is concerned instead it is worth to emphasize that by using the GC/TCD system, the highest sensitivity (slope = 71.4) and lower LOD was measured for formaldehyde, whereas the lowest sensitivity (slope = 0.29) and highest LOD was that of 2,5-Dimethylbenzaldehyde. When using GC/MS instead, the sensitivity and LODs were better for C_3_-C_5_ analytes than for aromatic carbonyls. Moreover, in this case, 2,5-dimethylbenzaldehyde was the CC that showed the calibration line with the lower slope and higher LOD. By looking at the data of GC/ECD it appeared that the highest sensitivity was attained for valeraldehyde and formaldehyde. These two compounds were also those characterized by the least LODs (0.003 ng/mL). The calibration plots for aromatic CCs had lower slopes and higher LODs (0.332–0.515), consistently with the MS results and with the fact that these molecules have the least tendency to act as electron donors.

### 3.3. Monitoring of CCs Concentration

The distribution of the outdoor and indoor levels of the 14 CCs between particulate and vapor phase measured during the sampling periods in 2018 are reported in Table 3 and Table 4.

The outdoor results presented in Table 3 show that the particle to vapor ratios (p/v) of C_1_-C_5_ CCs (i.e., columns FA to CRO) were in the range from 0.11 to 0.22, indicating that these CCs existed mainly in the vapor phase. The p/v ratios of C_6_ (HEX) and aromatic CCs (BENZ, m-TOL, and p,o-TOL, and 2,5-DMB), were always >1, indicating these carbonyls were present in the air predominantly as the particulate phase. The p/v ratios found here for the CCs considered are in agreement with the values reported by other authors [25,50].

The monitoring of CCs at the background site revealed that FA, AA, and acetone (AC) were the three most abundant carbonyls in the urban air of the area of Florence studied, with an average of 2.7, 2.9, and 1.4 µg/m^3^, respectively (Table 3).

In VE-dominated samples, the highest total CCs concentration was found in Viale Pieraccini. The more abundant pollutants found at the sampling sites of Viale Pieraccini were confirmed to be FA, AA, AC, and HEX, whereas BUT, CRO, and iso-VAL were the C_4_-C_5_ aldehydes more abundant in the samples collected at the site 3 of Via Caccini after FA, AA and AC. The other CCs measured in Viale Pieraccini had levels lower than 2.0 µg/m^3^, which accounted for less than 15% of the total amount of CCs quantified, and analogous trends were observed for the other sampling sites. The sites of Via delle Gore had an overall picture of CCs levels comparable to those of the roadside sites and of the Background site of sampling. Interestingly, although AC, BENZ, and VAL are not the major CCs, they can be regarded here as the more specific and meaningful markers of the pollution of the different sites. As can be seen from the data in Table 3, the Background site is characterized by low levels of AC, BENZ and VAL, whereas the Viale Pieraccini site n. 1 is the site with the highest levels of AC, BENZ, and VAL and is the more polluted sampling site. At the roadside sampling sites, again AA and AC were found to be the next two more abundant CCs after FA. The total quantified CCs at the four roadside locations were significantly higher than those measured at the background sampling site.

One possible explanation of these results is related to the number and fuel types of the vehicles passing near the sampling sites.

The traffic record of Viale Pieraccini (VE-dominated samples) showed an average of more than 5000 different types of LPG-, gasoline-, and diesel-fueled vehicles passing the site during the sampling events. At the roadside sampling locations, the vehicle numbers were higher than those counted on average at the background site. In addition, the roadside sampling sites were characterized by moderate traffic due mainly to heavy vehicles (diesel-fueled) engaged in the construction of the municipal tramway.

The concentration ratio of FA to AA (FA/AA) for both VE-dominated samples and roadside samples was evaluated. The FA/AA ratio was used widely to distinguish the sources in urban and rural areas [51]. Our data showed that the FA/AA ratio was close to 0.9–2.8 in the atmosphere in the urban areas, in agreement with the results reported by Ho and coworkers [52].

The above-mentioned evaluation approach of outdoor CCs concentration has been adopted to estimate the influence of outdoor pollution sources on indoor air quality by Scheepers, with sampling points inside and outside a hospital outer limits [53]. For healthcare center, outdoor sources of air pollution represent a potential threat to indoor air quality. We used this method to investigate the impact of CCs outdoor concentration on the pathology laboratories indoor air quality, where the large use of FA, as a fixing agent for surgical specimen, requires constant monitoring of the professional exposure, due to the carcinogenicity of this substance. To estimate the CCs’ exposure levels in these theaters, it’s fundamental to have an accurate evaluation of the background pollution, so we simultaneously monitored the CCs concentration in pathology laboratories’ critical areas (secretariat, laboratories during the slicing of previously surgical pathology specimens and offices) and in near outdoor sampling sites sources of air pollution.

Based on our results (Table 4), we could conclude that the influence of outdoor CCs concentration was minimal on the indoor levels of CCs, considering the high basal CCs’ concentration measured in the indoor air. This point was clear especially for FA, that is used in the pathology laboratories, and thus, it is present in the indoor air since it is released during the activities carried out in this health facility.

## 4. Conclusions

The combination of 2,4-DNPH coated annular denuders-filters with GC makes possible the quantitative detection of CCs in both very concentrated and diluted samples and it is a promising tool for environmental chemists interested in studying outdoor and indoor pollution. There are three main advantages of this method over traps filled with coated sorbents: i) Artifacts arising from CCs adsorbed on particulate matter can be avoided ii) higher flow rates can be used without loss of efficiency, iii) undesired reactions of CCs, 2,4-DNPH, and their derivatives with the sorbent can be minimized, because glass is used as inert support for the reagent.

## Figures and Tables

**Figure 1 ijerph-16-01969-f001:**
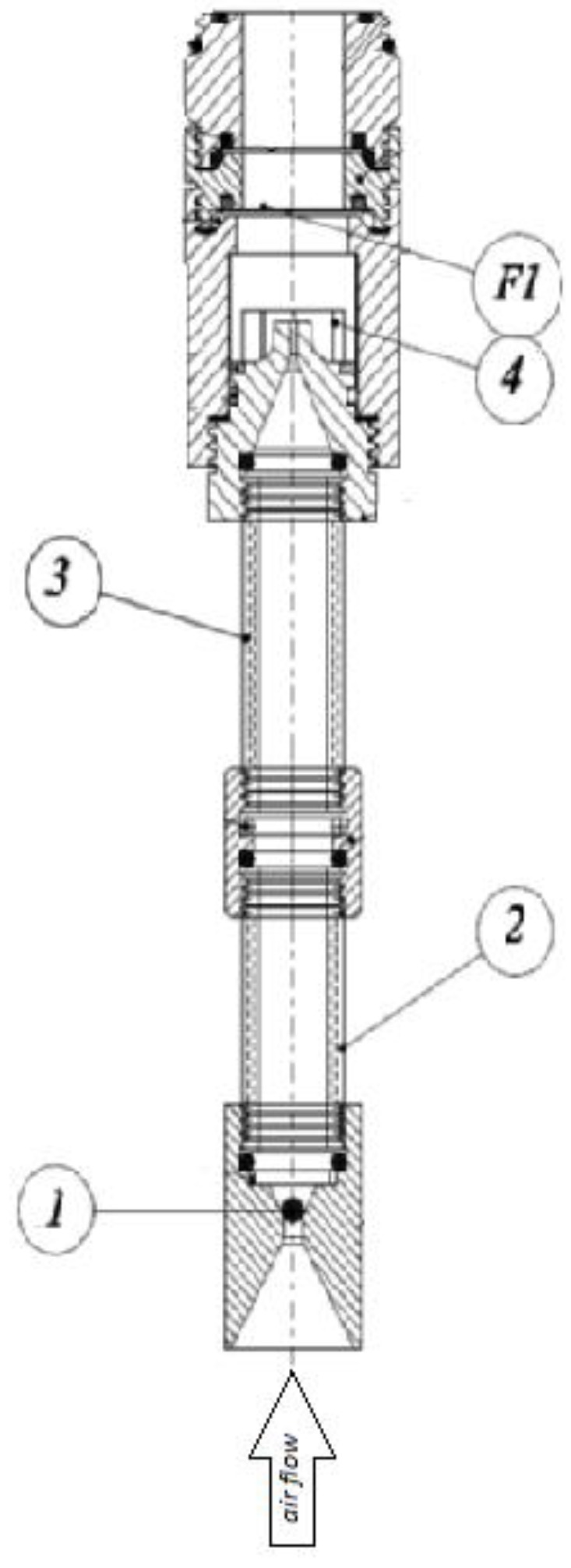
Denuder sampler. Teflon ball (1) which interrupts the passive airflow towards the denuders when the sampling pump is not in operation. Gases and particles enter the first denuder (2) and then in the second (3). Then, the air carrying the particulate passes into the impactor (4) where the PM_2.5_ fraction is collected by the filter F1 on which the concentration of the species in the particulate phase is subsequently determined.

**Figure 2 ijerph-16-01969-f002:**
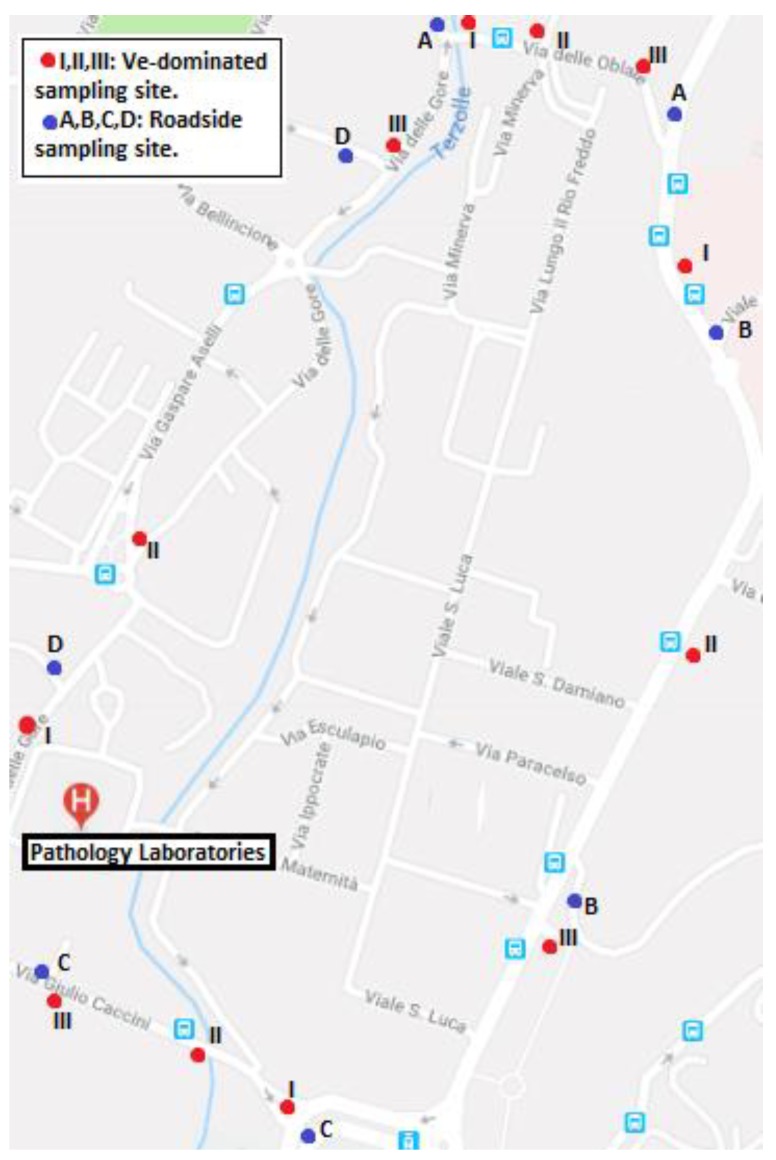
Map of the sampling sites around the Careggi University Hospital (roadside sampling sites situated in the same street are indicated with the same capital letter: E.g., A for the two roadside sampling sites at via delle Oblate).

**Table 1 ijerph-16-01969-t001:** Collection efficiency (E%) of two DNPH-coated denuders, tested with a carbonyl compounds (CCs) test solution.

Flowrate (L/min)	Time of Exposure (min)	FA			C_2_-C_3_ CCs		
		m_1_	m_2_	E%	m_1_	m_2_	E%
1.0	60	34.99	-	99.9	34.2	0.82	97.7

LEGEND: “-”: nondetectable; FA, formaldehyde; C_2_-C_3_ CCs, two- and three-carbon aldehydes.

**Table 2 ijerph-16-01969-t002:** Calibration parameters of the GC-ECD/TSD and MS systems for the analysis of carbonyl compounds (CCs).

CC	GC/TSD				GC/MS				GC/ECD			
	LOD (ng/mL)	m	y	SE	LOD (ng/mL)	m	y	SE	LOD (ng/mL)	m	y	SE
Formaldehyde	0.07	71.4	0.087	0.08	0.63	0.81	0.083	0.0846	0.003	154.3	0.076	0.078
Acetaldehyde	0.13	3.72	0.079	0.082	1.06	0.48	0.088	0.083	0.006	80.5	0.084	0.077
Acrolein	0.26	1.81	0.081	0.075	0.76	0.61	0.077	0.077	0.006	81.5	0.087	0.076
Hexaldehyde	0.76	0.63	0.083	0.076	0.83	0.55	0.081	0.072	0.031	15.58	0.079	0.082
Acetone	0.43	1.21	0.089	0.085	0.8	0.62	0.079	0.085	0.016	30	0.083	0.077
Propionaldehyde	0.13	3.83	0.079	0.087	0.43	1.14	0.088	0.076	0.006	84.5	0.083	0.086
Butyraldehyde	0.56	0.88	0.088	0.076	0.36	1.41	0.091	0.078	0.006	84	0.076	0.092
Isovaleraldehyde	0.6	0.87	0.082	0.092	0.63	0.77	0.086	0.076	0.013	41.31	0.091	0.088
Valeraldehyde	0.72	0.66	0.081	0.078	0.53	0.9	0.084	0.075	0.003	167	0.082	0.085
Crotonaldehyde	1.03	0.51	0.092	0.082	0.53	0.91	0.088	0.072	0.006	75	0.078	0.072
Benzaldehyde	1.06	0.48	0.077	0.092	0.96	0.49	0.075	0.081	0.332	1.44	0.083	0.076
m-Tolualdehyde	1.23	0.39	0.083	0.075	1.26	0.39	0.082	0.083	0.412	1.14	0.086	0.071
p-Tolualdehyde	1.28	0.37	0.073	0.086	1.24	0.41	0.087	0.084	0.423	1.2	0.082	0.087
o-Tolualdehyde	1.36	0.34	0.075	0.081	1.36	0.38	0.092	0.081	0.465	1.09	0.092	0.077
2,5-Dimethylbenzaldehyde	1.74	0.29	0.086	0.081	1.72	0.29	0.081	0.084	0.515	0.97	0.084	0.083

LEGEND: LOD, limit of detection; m, slope of the calibration curve; y, intercept on the y-axis; SE, standard error of the y-axis intercept of the calibration curve.

**Table 3 ijerph-16-01969-t003:** Outdoor carbonyl compounds (CCs) mean ± standard deviation concentration results measured in at least three replicate samples of particulate (p) and vapor (v) fractions.

CC		FA	AA	ACR	HEX	AC	PRP	BUT	Iso-VAL	VAL	CRO	BENZ	m-TOL	p- + o-TOL	2,5-DMB
Sampling Site															
Background (average)	p	0.3 ± 0.1	0.5 ± 0.2	0.03 ± 0.01	0.12 ± 0.03	0.2 ± 0.1	0.07 ± 0.02	0.05 ± 0.02	0.1 ± 0.02	0.04 ± 0.01	0.03 ± 0.01	0.3 ± 0.08	0.03 ± 0.01	0.12 ± 0.02	0.06 ± 0.01
	v	2.4 ± 0.6	2.4 ± 1.0	0.17 ± 0.05	0.09 ± 0.02	1.2 ± 0.4	0.33 ± 0.11	0.35 ± 0.07	0.6 ± 0.2	0.36 ± 0.11	0.17 ± 0.04	0.2 ± 0.03	0.02 ± 0.01	0.08 ± 0.01	0.04 ± 0.01
VE-d, V.delle Oblate, pt.I	p	1.5 ± 0.9	1.0 ± 0.5	0.04 ± 0.01	0.22 ± 0.05	0.5 ± 0.2	0.13 ± 0.02	0.15 ± 0.03	0.2 ± 0.04	0.1 ± 0.03	0.1 ± 0.03	0.9 ± 0.1	0.05 ± 0.02	0.15 ± 0.03	0.17 ± 0.03
	v	7.1 ± 1.3	4.8 ± 1.7	0.26 ± 0.07	0.13 ± 0.02	2.3 ± 0.3	0.67 ± 0.18	0.96 ± 0.17	1.1 ± 0.3	0.9 ± 0.2	0.8 ± 0.2	0.6 ± 0.08	0.03 ± 0.01	0.05 ± 0.01	0.13 ± 0.02
VE-d, V.delle Oblate, pt.II	p	1.6 ± 0.8	0.9 ± 0.5	0.07 ± 0.01	0.25 ± 0.06	0.6 ± 0.2	0.1 ± 0.15	0.1 ± 0.04	0.2 ± 0.04	0.2 ± 0.07	0.2 ± 0.04	1.0 ± 0.2	0.10 ± 0.03	0.19 ± 0.03	0.25 ± 0.05
	v	9.0 ± 1.8	4.3 ± 2.0	0.33 ± 0.06	0.17 ± 0.02	3.1 ± 0.7	0.5 ± 0.1	0.8 ± 0.20	1.2 ± 0.2	1.0 ± 0.3	1.0 ± 0.2	0.6 ± 0.1	0.08 ± 0.02	0.11 ± 0.02	0.15 ± 0.03
VE-d V.delle Oblate, pt.III	p	2.2 ± 0.8	0.6 ± 0.3	0.05 ± 0.02	0.22 ± 0.08	0.4 ± 0.2	0.11 ± 0.05	0.02 ± 0.01	0.2 ± 0.03	0.1 ± 0.03	0.1 ± 0.03	0.8 ± 0.2	0.16 ± 0.04	0.27 ± 0.04	0.31 ± 0.05
	v	10.1 ± 2.1	4.0 ± 1.3	0.35 ± 0.07	0.15 ± 0.04	3.0 ± 0.8	0.79 ± 0.19	1.0 ± 0.2	1.1 ± 0.1	1.1 ± 0.4	1.0 ± 0.3	0.7 ± 0.2	0.10 ± 0.03	0.13 ± 0.02	0.19 ± 0.03
VE-d, V.Pieraccini, pt.I	p	4.0 ± 1.2	2.0 ± 0.7	0.09 ± 0.02	0.45 ± 0.12	0.7 ± 0.1	0.15 ± 0.06	0.20 ± 0.06	0.25 ± 0.05	0.3 ± 0.09	0.2 ± 0.04	1.0 ± 0.4	0.45 ± 0.07	0.18 ± 0.02	0.39 ± 0.04
	v	18.7 ± 2.0	9.1 ± 2.0	0.71 ± 0.17	0.33 ± 0.10	4.5 ± 1.1	0.95 ± 0.19	1.1 ± 0.3	1.55 ± 0.32	1.5 ± 0.3	1.1 ± 0.2	0.8 ± 0.1	0.30 ± 0.05	0.12 ± 0.01	0.21 ± 0.03
VE-d, V.Pieraccini, pt.II	p	2.5 ± 1.0	1.4 ± 0.5	0.09 ± 0.03	0.41 ± 0.14	0.7 ± 0.2	0.1 ± 0.03	0.1 ± 0.04	0.2 ± 0.03	0.2 ± 0.02	0.15 ± 0.03	0.9 ± 0.2	0.12 ± 0.02	0.17 ± 0.02	0.27 ± 0.02
	v	13.1 ± 2.2	6.2 ± 1.3	0.41 ± 0.12	0.24 ± 0.09	3.4 ± 0.7	0.7 ± 0.1	0.9 ± 0.22	1.3 ± 0.27	1.2 ± 0.1	0.85 ± 0.14	0.6 ± 0.1	0.10 ± 0.01	0.13 ± 0.02	0.23 ± 0.02
VE-d, V.Pieraccini, pt.III	p	2.2 ± 1.2	0.9 ± 0.4	0.07 ± 0.02	0.38 ± 0.11	0.6 ± 0.2	0.08 ± 0.03	0.15 ± 0.03	0.1 ± 0.04	0.2 ± 0.04	0.1 ± 0.04	0.9 ± 0.3	0.19 ± 0.04	0.25 ± 0.04	0.38 ± 0.06
	v	13.1 ± 2.4	5.7 ± 1.2	0.33 ± 0.13	0.28 ± 0.09	2.9 ± 0.8	0.52 ± 0.12	0.65 ± 0.11	1.1 ± 0.3	1.4 ± 0.3	0.7 ± 0.2	0.7 ± 0.2	0.13 ± 0.02	0.15 ± 0.02	0.22 ± 0.05
VE-d, V.Caccini, pt.I	p	1.6 ± 0.7	0.6 ± 0.3	0.08 ± 0.01	0.26 ± 0.07	0.8 ± 0.2	0.06 ± 0.02	0.2 ± 0.08	0.1 ± 0.02	0.25 ± 0.06	0.1 ± 0.04	0.9 ± 0.3	0.22 ± 0.04	0.18 ± 0.03	0.17 ± 0.03
	v	9.0 ± 1.4	4.2 ± 1.1	0.42 ± 0.11	0.16 ± 0.04	3.5 ± 1.1	0.54 ± 0.11	1.0 ± 0.3	1.3 ± 0.3	1.15 ± 0.4	0.9 ± 0.3	0.6 ± 0.2	0.16 ± 0.05	0.12 ± 0.02	0.13 ± 0.02
VE-d, V.Caccini, pt.II	p	1.5 ± 0.7	0.5 ± 0.2	0.11 ± 0.03	0.23 ± 0.04	0.4 ± 0.2	0.05 ± 0.02	0.15 ± 0.03	0.2 ± 0.02	0.1 ± 0.03	0.15 ± 0.02	0.8 ± 0.2	0.16 ± 0.03	0.11 ± 0.01	0.23 ± 0.04
	v	7.4 ± 1.7	2.6 ± 0.6	0.49 ± 0.05	0.14 ± 0.03	1.9 ± 0.7	0.45 ± 0.13	0.85 ± 0.13	1.3 ± 0.2	1.1 ± 0.3	0.65 ± 0.14	0.5 ± 0.1	0.12 ± 0.03	0.09 ± 0.01	0.17 ± 0.02
VE-d, V.Caccini, pt.III	p	1.2 ± 1.0	0.5 ± 0.3	0.07 ± 0.02	0.25 ± 0.05	0.4 ± 0.2	0.1 ± 0.03	0.3 ± 0.08	0.1 ± 0.02	0.1 ± 0.03	0.2 ± 0.09	0.6 ± 0.1	0.07 ± 0.01	0.12 ± 0.02	0.35 ± 0.06
	v	5.3 ± 1.3	2.2 ± 0.4	0.43 ± 0.12	0.16 ± 0.03	1.8 ± 0.6	0.5 ± 0.1	1.5 ± 0.3	1.5 ± 0.3	0.8 ± 0.2	1.0 ± 0.2	0.5 ± 0.1	0.04 ± 0.01	0.08 ± 0.02	0.15 ± 0.02
VE-d, V. delle Gore, pt.I	p	1.7 ± 0.8	0.5 ± 0.3	0.03 ± 0.01	0.29 ± 0.07	0.3 ± 0.1	0.03 ± 0.01	0.07 ± 0.02	0.1 ± 0.03	0.1 ± 0.02	0.17 ± 0.03	0.5 ± 0.2	0.05 ± 0.02	0.16 ± 0.03	0.18 ± 0.03
	v	7.9 ± 2.0	3.1 ± 0.3	0.27 ± 0.07	0.22 ± 0.06	1.6 ± 0.4	0.17 ± 0.04	0.43 ± 0.10	0.8 ± 0.1	0.6 ± 0.18	0.63 ± 0.23	0.4 ± 0.1	0.03 ± 0.01	0.14 ± 0.03	0.12 ± 0.02
VE-d, V. delle Gore, pt.II	p	1.5 ± 0.3	0.7 ± 0.3	0.07 ± 0.01	0.23 ± 0.04	0.3 ± 0.1	0.07 ± 0.02	0.08 ± 0.02	0.1 ± 0.04	0.1 ± 0.04	0.1 ± 0.04	0.5 ± 0.08	0.05 ± 0.01	0.12 ± 0.02	0.11 ± 0.01
	v	7.3 ± 2.1	3.2 ± 1.0	0.33 ± 0.09	0.16 ± 0.03	1.6 ± 0.5	0.43 ± 0.09	0.52 ± 0.14	0.8 ± 0.25	0.7 ± 0.21	1.0 ± 0.3	0.3 ± 0.1	0.03 ± 0.02	0.08 ± 0.01	0.09 ± 0.01
VE-d, V. delle Gore, pt.III	p	1.2 ± 0.4	0.8 ± 0.3	0.08 ± 0.02	0.35 ± 0.11	0.3 ± 0.1	0.08 ± 0.02	0.1 ± 0.05	0.2 ± 0.04	0.13 ± 0.03	0.1 ± 0.04	0.5 ± 0.1	0.06 ± 0.03	0.13 ± 0.02	0.18 ± 0.02
	v	7.6 ± 1.8	4.4 ± 1.3	0.42 ± 0.14	0.26 ± 0.09	1.9 ± 0.7	0.42 ± 0.11	0.7 ± 0.1	0.9 ± 0.2	0.67 ± 0.13	1.0 ± 0.2	0.3 ± 0.04	0.04 ± 0.02	0.07 ± 0.01	0.12 ± 0.01
Roadside, A (average)	p	2.2 ± 0.9	1.0 ± 0.4	0.07 ± 0.03	0.24 ± 0.07	0.2 ± 0.1	0.05 ± 0.02	0.1 ± 0.03	0.1 ± 0.04	0.05 ± 0.01	0.11 ± 0.02	0.4 ± 0.1	0.05 ± 0.02	0.11 ± 0.02	0.11 ± 0.02
	v	10.0 ± 2.4	4.6 ± 1.2	0.33 ± 0.11	0.14 ± 0.05	1.4 ± 0.6	0.45 ± 0.10	0.6 ± 0.14	0.8 ± 0.15	0.55 ± 0.11	0.59 ± 0.11	0.2 ± 0.03	0.04 ± 0.01	0.09 ± 0.02	0.09 ± 0.01
Roadside, B (average)	p	1.8 ± 0.3	0.7 ± 0.2	0.05 ± 0.03	0.24 ± 0.08	0.2 ± 0.1	0.06 ± 0.02	0.1 ± 0.02	0.08 ± 0.03	0.1 ± 0.04	0.08 ± 0.03	0.4 ± 0.1	0.04 ± 0.02	0.12 ± 0.03	0.16 ± 0.02
	v	8.5 ± 1.6	3.4 ± 1.3	0.25 ± 0.09	0.17 ± 0.04	1.4 ± 0.4	0.34 ± 0.07	0.5 ± 0.07	0.92 ± 0.23	0.6 ± 0.13	0.42 ± 0.12	0.3 ± 0.08	0.03 ± 0.02	0.08 ± 0.02	0.14 ± 0.02
Roadside, C (average)	p	1.0 ± 0.4	0.6 ± 0.2	0.06 ± 0.02	0.20 ± 0.08	0.3 ± 0.1	0.03 ± 0.01	0.11 ± 0.02	0.1 ± 0.03	0.04 ± 0.02	0.08 ± 0.02	0.4 ± 0.1	0.05 ± 0.01	0.17 ± 0.01	0.17 ± 0.03
	v	6.3 ± 2.2	3.0 ± 1.1	0.44 ± 0.10	0.15 ± 0.04	1.4 ± 0.5	0.27 ± 0.08	0.49 ± 0.12	0.77 ± 0.23	0.54 ± 0.14	0.52 ± 0.10	0.3 ± 0.03	0.03 ± 0.01	0.13 ± 0.02	0.13 ± 0.01
Roadside, D (average)	p	1.0 ± 0.5	0.6 ± 0.3	0.03 ± 0.01	0.21 ± 0.06	0.2 ± 0.1	0.09 ± 0.02	0.06 ± 0.01	0.05 ± 0.01	0.05 ± 0.02	0.1 ± 0.03	0.4 ± 0.1	0.05 ± 0.02	0.11 ± 0.02	0.18 ± 0.02
	v	6.2 ± 1.8	3.2 ± 1.3	0.27 ± 0.06	0.15 ± 0.03	1.3 ± 0.3	0.41 ± 0.13	0.44 ± 0.13	0.55 ± 0.12	0.55 ± 0.16	0.7 ± 0.21	0.2 ± 0.07	0.04 ± 0.02	0.09 ± 0.01	0.12 ± 0.01

LEGEND: FA (formaldehyde), AA (acetaldehyde), ACR (acrolein), HEX (hexaldehyde), AC (acetone), PRP (propionaldehyde), BUT (butyraldehyde), iso-VAL (iso-valeraldehyde), VAL (valeraldehyde), CRO (crotonaldehyde), BENZ (benzaldehyde), m-TOL (m-tolualdehyde), p- + o-TOL (p-+o- tolualdehyde), 2,5-DMB (2,5-dimethylbenzoaldehyde); VE-d, vehicule dominated site; pt.I, II, III, sampling positions selected within the site.

**Table 4 ijerph-16-01969-t004:** Pathology laboratories facility’s indoor carbonyl compounds (CCs) average concentration results of at least three replicate measurements of particulate (p) and vapor (v) fractions.

CC		FA	AA	ACR	HEX	AC	PRP	BUT	Iso-VAL	VAL	CRO	BENZ	m-TOL	p-/o-TOL	2,5-DMB
Sampling Site															
Secretariat	p	1.4 ± 0.2	0.7 ± 0.2	0.08 ± 0.03	0.17 ± 0.1	0.3 ± 0.1	0.07 ± 0.08	0.14 ± 0.03	0.3 ± 0.1	0.2 ± 0.1	0.08 ± 0.04	0.8 ± 0.2	0.03 ± 0.01	0.17 ± 0.02	0.37 ± 0.15
	v	11.2 ± 0.4	4.0 ± 0.5	0.42 ± 0.05	0.12 ± 0.1	1.4 ± 0.4	0.43 ± 0.10	1.1 ± 0.3	1.5 ± 0.3	1.6 ± 0.3	0.82 ± 0.15	0.6 ± 0.1	0.02 ± 0.01	0.13 ± 0.02	0.13 ± 0.02
Pathology Lab.	p	1.6 ± 0.4	0.8 ± 0.3	0.05 ± 0.02	0.20 ± 0.07	0.7 ± 0.3	0.11 ± 0.08	0.15 ± 0.07	0.2 ± 0.1	0.2 ± 0.1	0.3 ± 0.1	0.7 ± 0.2	0.13 ± 0.03	0.27 ± 0.1	0.18 ± 0.05
	v	12.3 ± 0.6	4.4 ± 0.7	0.45 ± 0.06	0.12 ± 0.06	2.2 ± 0.5	0.79 ± 0.11	0.85 ± 0.1	1.0 ± 0.3	1.2 ± 0.3	1.3 ± 0.3	0.5 ± 0.1	0.09 ± 0.02	0.13 ± 0.08	0.12 ± 0.04
Offices	p	1.2 ± 0.3	0.5 ± 0.2	0.07 ± 0.02	0.15 ± 0.1	0.7 ± 0.2	0.05 ± 0.03	0.18 ± 0.06	0.17 ± 0.1	0.11 ± 0.1	0.09 ± 0.02	0.5 ± 0.2	0.10 ± 0.02	0.12 ± 0.06	0.16 ± 0.04
	v	8.7 ± 0.8	3.1 ± 0.4	0.33 ± 0.04	0.12 ± 0.07	1.7 ± 0.3	0.45 ± 0.09	0.82 ± 0.1	0.83 ± 0.2	0.79 ± 0.1	0.71 ± 0.2	0.4 ± 0.1	0.07 ± 0.03	0.08 ± 0.03	0.14 ± 0.03

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
