# Peer review of "Monitoring of Air-Dispersed Formaldehyde and Carbonyl Compounds as Vapors and Adsorbed on Particulate Matter by Denuder-Filter Sampling and Gas Chromatographic Analysis"

_ijerph, 2019, doi:10.3390/ijerph16111969_

Reviewer 1 Report

This article describes a sampling method used for analyzing and measuring carbonyl compounds emissions. The advantages of this method (e.g., its high accuracy) over similar methods were clearly elaborated in the introduction. Methodology section also contains sufficient amount of information. My only concern is organization of the results and discussion section. I understand that the aim of the article is introducing a new sampling method, but I believe that the results and discussion contains too much information and is less focused on the discussion of the results. More importantly, the discussion of data are not well organized. For instance the data in Table 1 discussed in two spots. I recommend this paper for publication after addressing the above-mentioned comments as well as those minor ones listed as below

Line 88: please change “analyzed” to “analyze”.

Line 93: The following sentence is currently vague. Please make it clear: “As the conventional methods using 2,4-DNPH impregnated cartridge to collect CCs cannot differentiate gaseous and particulate carbonyls, the denuder-filter sampling technique is explored to collect them separately”.

Line 129: Please remove “s” in “captures”.

Line 366: Please elaborate more on how the experimental results agree well with calculations.

Line 367: “Collection efficiencies for the other CCs were, however, much lower than those expected.” Collection efficiency should be included in the table.

In general more explanation on data processing and more discussion is needed for Table 1. The First row and column for this table should be labeled.

m, y, and SE are needed to be defined in the table 2.

Line 409: “The calibrations parameters (LOD, slope, intercept and standard error) obtained by using the GC-ECD/TSD and MS systems in the determination of studied analytes were compared in Table 2” More discussion on comparing data and trends in Table 2 is needed.  

Line 428: “The results show that the particle/gas ratios of C1-C5 CCs were in the range from 0.11 to 0.22, indicating that these CCs existed mainly in vapour phase” I cannot see any information about particle on Table 1. How did you obtain particle/gas ratio?

Line 433: “Background monitoring revealed that FA, AA, and acetone (AC) were the three most abundant carbonyls, with an average of 2.7±2.0, 2.9±0.7, and 1.4±1.1 μg/m3, respectively.” I cannot understand why the results in Table 1 is discussed here again? It is better all the data in each Table to be discussed completely in one place. Although I understand it is bringing under a new title, I believe it is more relevant to have it before in one place.

I recommend that the two figures in Fig. 3 to be brought in a column not in a row. Because in this way you can have it larger as currently the details cannot be clearly seen.

Other than the contribution of the developed method, you need to bring the main conclusions from your measurements, e.g., the CCs with the highest concentrations, etc.

Author Response

Response to reviewer 1 comments

Presentation of the results can be improved (R1).

Conclusions can be supported better by results (R1).

The presentation of the results and the conclusions have been rewritten and revised adding more data and expanding the section dedicated to the results discussion.

Additional changes made include the introduction of new tables, the revision of the section regarding the data analysis (elimination of sections 2.7 and 3.4), and a slight modification of the references list in the bibliography.

This article describes a sampling method used for analyzing and measuring carbonyl compounds emissions. The advantages of this method (e.g., its high accuracy) over similar methods were clearly elaborated in the introduction. Methodology section also contains sufficient amount of information. My only concern is organization of the results and discussion section. I understand that the aim of the article is introducing a new sampling method, but I believe that the results and discussion contains too much information and is less focused on the discussion of the results. More importantly, the discussion of data are not well organized. For instance the data in Table 1 discussed in two spots. I recommend this paper for publication after addressing the above-mentioned comments as well as those minor ones listed as below

The sections of the manuscript regarding the results and their discussion have been rewritten almost entirely taking into consideration the criticism of the Reviewers. The discussion has been reorganized in order to be more consistent and to discuss in single paragraphs each aspect worth of note of the results presented.

The revision of the manuscript included rewriting Section 3, the introduction of 3 additional new tables (Tables 1, 2, and 3), the elimination of paragraphs describing the PCA of the data, and detailed discussion of the data presented.

R1_3) Line 88: please change “analyzed” to “analyze”.

Change introduced in the text (line 90).

R1_4) Line 93: The following sentence is currently vague. Please make it clear: “As the conventional methods using 2,4-DNPH impregnated cartridge to collect CCs cannot differentiate gaseous and particulate carbonyls, the denuder-filter sampling technique is explored to collect them separately”.

The sentence was reworded to make it more readable and clear. The revised sentence (lines 96-99) is “Conventional methods using cartridges impregnated with 2,4-DNPH are not able to differentiate the two sample fractions represented by the air-dispersed CCs in vapor phase and those adsorbed on particulate matter. Thanks to the use of the denuder-filter sampling technique instead, it is possible to collect the two fractions separately [24,25].”

R1_5) Line 129: Please remove “s” in “captures”.

Change introduced in the text (line 133).

R1_6) Line 366: Please elaborate more on how the experimental results agree well with calculations.

R1_7) Line 367: “Collection efficiencies for the other CCs were, however, much lower than those expected.” Collection efficiency should be included in the table.

The experimental results have been more widely discussed in the text. New tables have also been added in order to better illustrate our results, as also requested by the other reviewers. The discussion regarding the agreement between experimental results and calculations is now written at lines 391-401 and summarized in Table 3 “Collection efficiency (E%) of two DNPH-coated denuders, tested with a Carbonyl Compounds (CCs) test solution”.

R1_8) In general more explanation on data processing and more discussion is needed for Table 1.

In the revised manuscript, we expanded the section regarding the discussion of the data and modified entirely the part concerning data processing. Tables 1-2 now describe in more detail our results and Section 3.3 has been almost entirely rewritten.

R1_9) The First row and column for this table should be labeled.

The first row and column of former Table 1 have been now labeled. Table 1 has been revised following also the suggestions made by the Reviewer 1 (here and above).

R1_10) m, y, and SE are needed to be defined in the table 2.

The calibration lines parameters have been now defined in the Legend to former Table 2 now renamed as Table 5. Also, the Table caption has been modified accordingly, and the first column has been labeled.

R1_11) Line 409: “The calibrations parameters (LOD, slope, intercept and standard error) obtained by using the GC-ECD/TSD and MS systems in the determination of studied analytes were compared in Table 2” More discussion on comparing data and trends in Table 2 is needed.

One paragraph discussing the data information contained in former Table 2, now Table 4, has been added at lines 450-463.

R1_12) Line 428: “The results show that the particle/gas ratios of C1-C5 CCs were in the range from 0.11 to 0.22, indicating that these CCs existed mainly in vapour phase” I cannot see any information about particle on Table 1. How did you obtain particle/gas ratio?

In the revised manuscript we provided two new separate tables for particle-adsorbed and vapor-phase CCs to provide the requested information regarding the particle/vapor ratio discussed in the text (lines 475-480).

R1_13) Line 433: “Background monitoring revealed that FA, AA, and acetone (AC) were the three most abundant carbonyls, with an average of 2.7±2.0, 2.9±0.7, and 1.4±1.1 μg/m3, respectively.”

I cannot understand why the results in Table 1 is discussed here again? It is better all the data in each Table to be discussed completely in one place. Although I understand it is bringing under a new title, I believe it is more relevant to have it before in one place.

The discussion of the results of Table 1 is not presented in Section 3.3 only.

R1_14) I recommend that the two figures in Fig. 3 to be brought in a column not in a row. Because in this way you can have it larger as currently the details cannot be clearly seen.

The Figures of former figure 3 have been eliminated following a careful examination of the recommendations of the Reviewers.

R1_15) Other than the contribution of the developed method, you need to bring the main conclusions from your measurements, e.g., the CCs with the highest concentrations, etc.-->

We included additional conclusions in Section 3.3 to illustrate how the results presented have been of impact in ongoing parallel studies on indoor pollution by CCs in our Hospital (lines 498-523).

Reviewer 2 Report

Broad Comments

This research provides an original, well-designed, and validated instrumental analysis method for determining carbonyl compounds in atmospheric samples.  The sampling method developed will be able to effectively provide measurements of carbonyls that have important implications for human health.

Numerous grammatical errors exist throughout the paper.  Specific errors are listed below.

Significant clarification is needed throughout the results and discussion section as to which measurements are being discussed in each section.  In particular, it is unclear what values are shown in Table 1.  The measurements for the vapor and aerosol phase concentrations of CCs is either missing or not clearly presented.  The efficiency study results for both vapor and particle CC concentration is also missing or not clearly presented.

The analysis of the experimental measurements is limited and the inclusion of the principle component analysis needs additional justification.  Please see specific comments for more detail.

Specific Comments

There are missing subscripts or superscripts throughout paper, in particular when ÎĽg m-3, PM2.5, CO2, cm2 are used and for the measurements in lines 359.

Multiple grammatical or spelling errors were present.  A partial list includes line 25 (“determination”), line 32 (“to detect”), line 168 (“weightings”), line 199 (“opening”), line 247 (“2.5 PM”), line 251 (“points”), line 325 (“Backgroud” in Table 1), line 409 and caption for Table 2 (“calibrations”).

Some structural errors in sentences.  The wording was confusing in lines 363-365.

There were some punctuation mistakes throughout the paper.  An extra period was included in line 215.  There are missing commas throughout.  A comma was missing in sentences in lines 313-315 which makes meaning of sentence unclear.

The statement made in lines 75-77 is not clear that it related to micrometer aggregates from diesel exhaust.

The time period for the CC exposure limits should be provided in lines 83-86.

Some parts of the method are unclear.  Line 149 indicated 5mL of coating solution was used but in line 150, 1 mL of coating solution was listed. 

Figure 1 shows 2 denuders in the sampling train, but methods (line 153) mentions that there are 3 denuders in the sampling train.  This needs to be clarified.  If a backup denuder was used, it would be nice to see the concentrations measured on this denuder within the text or in supplemental information.

What was the temperature program used for the GC separation?  (lines 201-207)

What is the difference between the VE-dominated and roadside sites?  What are the expected emission sources for the roadside sites? Provide a justification for creating these two sampling site categories (line 264).

Please include a site legend for the map in Figure 2.  The resolution of the map image should also be improved.

I don’t quite understand how you are assessing indoor air quality.  What do you mean by “sampling points inside and outside the hospital outer limits”?  These should be clearly labeled on the map in Figure 2.  Clarification is needed on the application of this work to indoor measurements. 

The site names listed in the Table 1 and Table 2 should clearly correspond with the sites shown in the map in Figure 2.

In Table 1, what is the value shown after the measurement (standard deviation, 95% confidence interval, etc.)?  It is also not clear if the values in the table are CC vapors from the denuder measurements or the adsorbed CCs on aerosol from the filter measurements, or a sum of both measurements.  A more descriptive caption is needed as well as a more clear description within the text.  Lines 366-368 indicate that Table 1 shows the collection efficiency measurements, but I do not find this clear.  The collection efficiency data need to be presented in a clear way in a table or figure separate from the denuder and filter measurements.

The presentation of the data in Table 1 would be improved through the use of a figure.  Either within the text of in the Supplemental Information, measurements from the two denuders and filters should be shown.

In lines 366-367, I’m not sure how you are able to compare the experimental data in Table 1 with calculated values of FA.  I do not see the values for calculated FA or any statistics comparing those values.

Please provide some analysis of your measurements and humidity to back up your statement in lines 373-374 and again in lines 378-379.  A summary of humidity and perhaps other meteorological measurements should be included.

Currently, the PCA does not significantly enhance this study.  A more clear description of both panels presented in Figure 3 is needed.  A more in depth analysis of the PCA results and a description of how it adds additional and unique information to this study is needed to justify including PCA.

Author Response

Response to reviewer 2 comments

Reviewer 2

Broad Comments

This research provides an original, well-designed, and validated instrumental analysis method for determining carbonyl compounds in atmospheric samples. The sampling method developed will be able to effectively provide measurements of carbonyls that have important implications for human health.

Numerous grammatical errors exist throughout the paper. Specific errors are listed below.

Significant clarification is needed throughout the results and discussion section as to which measurements are being discussed in each section. In particular, it is unclear what values are shown in Table 1.  The measurements for the vapor and aerosol phase concentrations of CCs is either missing or not clearly presented.  The efficiency study results for both vapor and particle CC concentration is also missing or not clearly presented.

The analysis of the experimental measurements is limited and the inclusion of the principle component analysis needs additional justification.  Please see specific comments for more detail.

R2_G1) Numerous grammatical errors exist throughout the paper. Specific errors are listed below.

The text has been thoroughly revised in order to amend all grammatical errors, mistyping errors, punctuation, missing subscripts/superscripts.

The results and discussion section has been entirely rewritten following the recommendations of the Reviewer. We hope that the clarification made copes with the Reviewer comment.

Table 1 has been deleted and more data have been added in new tables (1 and 2) in order to clarify the results obtained regarding the amount of CCs distributed in the vapor and in the particulate phases.

The efficiency study has been expanded and data have now been presented in anew Table (Table 3) to illustrate the efficiency of the denuder system used.

The principal component analysis of the data has been eliminated from the revised manuscript.

R2_7) There are missing subscripts or superscripts throughout paper, in particular when ÎĽg m-3, PM2.5, CO2, cm2 are used and for the measurements in lines 359.

All subscripts and superscripts have been corrected in the revised text.

R2_8) Multiple grammatical or spelling errors were present. A partial list includes

line 25 (“determination”),

the word “determination” has been changed as “measurement of the concentrations of 14 CCs”.

line 32 (“to detect”),

The entire sentence was redundant here and has been deleted.

line 168 (“weightings”),

the word “weightings” has been changed as “weighings” (line 172).

line 199 (“opening”),

the sentence was rewritten entirely. Now it reads as follows (lines 204-210). “To inject large amounts of ethyl acetate containing the analytes into the 1078/1079 PTV inlet, the solvent must be eliminated instantly while retaining all the target analytes in the liner. To achieve this effect during injection, the split vent valve must remain open while a large stream of carrier gas removes the solvent. After complete removal of the solvent, the split vent valve is closed and the inlet temperature increased. The analytes inside the liner are then rapidly vaporized and transferred into the column. The split vent is then opened again to remove the remaining traces of solvent vapor from the liner”.

line 247 (“2.5 PM”),

“2.5PM” has now been replaced by “PM2.5” in all occurrences throughout the manuscript.

line 251 (“points”),

“points” is now replaced by “Five-point” (line 265).

line 325 (“Backgroud” in Table 1),

“Backgroud” in former Table 1 now has been replaced by “Background”.

line 409 and caption for Table 2 (“calibrations”).

“Calibrations” has now been replaced by “Calibration” in the text (line 447) and in the caption to Table 4.

R2_9) Some structural errors in sentences. The wording was confusing in lines 363-365.

The sentences in lines 363-365 of the original manuscript have been reworded as follows at lines 391-394. “These values are much lower than those of gaseous species. Since the equation describing the fractional penetration of a monodisperse aerosol through an annular denuder has the same form as that of a gas, we conclude that particle deposition is negligible during sampling of aerosol at concentrations environmentally relevant”. We hope that the revised sentences cope with the Reviewer’s comment.

R2_10) There were some punctuation mistakes throughout the paper.

An extra period was included in line 215. There are missing commas throughout.

We would like to thank the Reviewer. Several mistyping errors were found in the text here. The sentences of the paragraph mentioned were thoroughly revised as follows (lines 228-233). “For the generation of air samples containing known concentrations of CCs that could model convincingly actual air samples, we made use of the static system proposed by Pieraccini et al. with modifications [40]. A volume of 5µL of aqueous solution of known CCs concentration (0.4–51.2 μg/μL) was injected using a 10μL GC syringe into the injector port at 200 °C of a modified Adsorbent Tube Injector System (ATIS, Cat. No. 28521, Supelco, Bellefonte, US) and collected into 100 L Tedlar sampling bags (Cat. No. KB3-50, Sensydine).”

R2_11) A comma was missing in sentences in lines 313-315 which makes meaning of sentence unclear.

The sentence was revised as follows (lines 325-327). “In 1983 annular denuders were applied for accurate measurements of gas-particle distribution of volatile compounds [43]. Since 1983, annular denuders have been widely used for sampling ambient air pollutants.”

R2_12) The statement made in lines 75-77 is not clear that it related to micrometer aggregates from diesel exhaust.

The sentence quoted by the Reviewer was revised as follows (lines 75-79). “Moskal et al. [16] reported that approximately 20% of the amount of aerosol inhaled through the nose penetrate the respiratory system when the aerosol is made by fractal-like aggregates having dimensions in the range from 1.7-2.1 µm and radius of gyration of 0.24-0.36 µm. A weak dependence of deposition efficiency on fractal dimensions and the radius of gyration values of the aerosol particles was also documented.”

R2_13) The time period for the CC exposure limits should be provided in lines 83-86.

The time period has been added in the text (lines 87-88).

R2_14) Some parts of the method are unclear. Line 149 indicated 5mL of coating solution was used but in line 150, 1 mL of coating solution was listed

The denuder was coated with 1 ml of the solution. This point has been also clarified in the revised text (lines 151-156). On a daily basis, we used to prepare 5 mL of stock solution and from this solution we used to take 1 mL to coat the denuder.

R2_15) Figure 1 shows 2 denuders in the sampling train, but methods (line 153) mentions that there are 3 denuders in the sampling train. This needs to be clarified. If a backup denuder was used, it would be nice to see the concentrations measured on this denuder within the text or in supplemental information

We apologize for the misunderstanding, there was a mistake in the text. Only two denuders have been used. The sentence has been rewritten to clarify this issue in the text at lines 158-160.

R2_16) What was the temperature program used for the GC separation?  (lines 201-207)

The temperature program used for the GC separation has been added in the text at lines 221-222.

R2_17) What is the difference between the VE-dominated and roadside sites? What are the expected emission sources for the roadside sites? Provide a justification for creating these two sampling site categories (line 264).

In the revised text we expanded the description of the sampling sites in order to clarify the differences between them (lines 277-281). The results have been commented taking into consideration the different picture of pollution found in the different sites and the possible relation with the number and fueling of the vehicles transiting the sites during the monitoring campaign (lines 499-506).

R2_18) Please include a site legend for the map in Figure 2. The resolution of the map image should also be improved.

A legend has been included for the map of Figure 2. The resolution of the image has been enhanced.

R2_19) I don’t quite understand how you are assessing indoor air quality. What do you mean by “sampling points inside and outside the hospital outer limits”? These should be clearly labeled on the map in Figure 2. Clarification is needed on the application of this work to indoor measurements.

We thank the Reviewer for this observation. In Section 3.3 we added a paragraph to answer the question raised by the Reviewer. We have clarified this part by adding our study on the indoor air quality in Careggi University Hospital (lines 511-526). Indoor air quality in the Hospital Pathology department has been evaluated during the same sampling period of the outdoor air evaluation reported in the present study.

R2_20) The site names listed in the Table 1 and Table 2 should clearly correspond with the sites shown in the map in Figure 2.

The four sampling locations (corresponding to the four streets of Florence mentioned in the text), were studied by putting more than one sampling apparatus for each street in order to control as accurately as possible all variations due to the outdoor environmental monitoring. The names of the samples therefore are made by the name of the sampling site followed by the number of the replicate collected.

R2_21) In Table 1, what is the value shown after the measurement (standard deviation, 95% confidence interval, etc.)?

The revised Table 1 now includes a caption that clarifies that the measurements are expressed in µg/m3 as mean±standard deviation.

R2_22) It is also not clear if the values in the table are CC vapors from the denuder measurements or the adsorbed CCs on aerosol from the filter measurements, or a sum of both measurements. A more descriptive caption is needed as well as a more clear description within the text.

R2_23) Lines 366-368 indicate that Table 1 shows the collection efficiency measurements, but I do not find this clear. The collection efficiency data need to be presented in a clear way in a table or figure separate from the denuder and filter measurements.

R2_24) The presentation of the data in Table 1 would be improved through the use of a figure. Either within the text of in the Supplemental Information, measurements from the two denuders and filters should be shown.

The data of former Table 1, now deleted and better explained in new Table 1 , have been implemented by additional data reported in two new tables (Tables 1 and 2). The results section 3.3 has been rewritten and we hope that the description of the efficiency of the denuder system now may represent a sufficient answer to the reviewer criticisms.

R2_25) In lines 366-367, I’m not sure how you are able to compare the experimental data in Table 1 with calculated values of FA. I do not see the values for calculated FA or any statistics comparing those values.

The comparison of the computed and measured data for FA as well as for C2-C3 CCs is now presented in Table 1 and commented in the text (lines 394-400). Considering that information provided by the raw data was sufficient, we did not include a statistical analysis.

R2_26) Please provide some analysis of your measurements and humidity to back up your statement in lines 373-374 and again in lines 378-379. A summary of humidity and perhaps other meteorological measurements should be included.

Data on RH% ad a brief comment on the robustness of the denuder under RH% varying from 50 to 90% have been included in the revised manuscript (lines 411-417).

R2_27) Currently, the PCA does not significantly enhance this study. A more clear description of both panels presented in Figure 3 is needed. A more in depth analysis of the PCA results and a description of how it adds additional and unique information to this study is needed to justify including PCA.

The PCA of data of Table 1 has now been eliminated from the revised manuscript, following the suggestions made by Reviewers.

Reviewer 3 Report

This study aims at sampling and monitoring gaseous and particulate carbonyl compounds simultaneously. However, the flaws of this manuscript are obvious. The result section is not easy to understand, and the conclusion section is not supported by the results. Also, the English needs to be polished extensively, some sentences are difficult to understand.

Some specific comments:

1.     Line 22: first sentence in the abstract, carbonyl compounds are not only combustion products but can be formed in the atmosphere (the authors mentioned this in the Introduction, i.e., secondary formation). This sentence needs to be revised accurately.

2.     The authors discussed a lot of table 1 in the result section. However, I cannot get that much information in table 1. I only see a concentration list of 14 CCs and FA/AA at different sampling sites and background. Maybe the authors did a lot of calculation but not showing in the manuscript. Some examples

Line 366, “By looking at the results of Table 1, we can say that experimental data agree well with those calculated from Eq. 4f”. How can readers understand the experimental data agree well with the calculated ones?

Line 428, “The results show that the particle/gas ratios of C1-C5 CCs were in the range from 0.11 to 0.22, indicating that these CCs existed mainly in vapour phase;” Which value is particle and gas concentration? It’s not explained in table 1.

3.     In section 3.1, it’s about evaluating the performance of denuder-filter sampler. But the authors used several paragraphs discussing the choice of LC/MS/MS or GC-ECD/TSD. These discussions are not related to this section. The authors may make a new section for a comparison of using different instrumentations.

4.     Again, the PCA result should not put in the section “Pictures of the CCs concentrations”, it should be divided into an independent section. BTW, line 479, “This section may be divided by subheadings. It should provide a concise and precise description of the experimental results, their interpretation as well as the experimental conclusions that can be drawn.” Is this sentence related to the PCA analysis discussion?

5.     The quality of Figure 2 is bad, please improve it.

Author Response

Response to reviewer 3 comments

Reviewer 3

This study aims at sampling and monitoring gaseous and particulate carbonyl compounds simultaneously. However, the flaws of this manuscript are obvious. The result section is not easy to understand, and the conclusion section is not supported by the results. Also, the English needs to be polished extensively, some sentences are difficult to understand.

We have revised extensively the text of the manuscript taking in consideration the criticisms expressed by Reviewer 3. The Results section has been entirely rewritten and the Conclusion section now should appear better supported by the results. The English was carefully revised throughout the entire text.

Some specific comments:

R3_1.     Line 22: first sentence in the abstract, carbonyl compounds are not only combustion products but can be formed in the atmosphere (the authors mentioned this in the Introduction, i.e., secondary formation). This sentence needs to be revised accurately.

The first sentence of the abstract has been revised deleting the word “combustion”.

R3_2.     The authors discussed a lot of table 1 in the result section. However, I cannot get that much information in table 1. I only see a concentration list of 14 CCs and FA/AA at different sampling sites and background. Maybe the authors did a lot of calculation but not showing in the manuscript. Some examples

The Results section has been entirely rewritten. Please see also the answers to the other reviewers.

R3_Line 366, “By looking at the results of Table 1, we can say that experimental data agree well with those calculated from Eq. 4f”. How can readers understand the experimental data agree well with the calculated ones?

We apologize for the misleading error made while writing the first draft of our manuscript. A new table (Table 3) and a short comment to the data (lines 394-400) have been introduced in the revised manuscript in order to clarify the issue raised by the Reviewer.

R3_Line 428, “The results show that the particle/gas ratios of C1-C5 CCs were in the range from 0.11 to 0.22, indicating that these CCs existed mainly in vapour phase;” Which value is particle and gas concentration? It’s not explained in table 1.

We agree with the Reviewer. A mistake was made here because we did not present the data of the distribution between the particulate and the vapor phases of the pollutants investigated. In the revised manuscript we thus present the missing data in two new Tables (Tables 1 and 2) that document the observations quoted by the Reviewer in this comment.

R3_3.     In section 3.1, it’s about evaluating the performance of denuder-filter sampler. But the authors used several paragraphs discussing the choice of LC/MS/MS or GC-ECD/TSD. These discussions are not related to this section. The authors may make a new section for a comparison of using different instrumentations.

We agree with the Reviewer comment. A new section (Section 3.2) has now been dedicated to the discussion of the selection of the more suitable combination of techniques for monitoring CCs in ambient air.

R3_4.     Again, the PCA result should not put in the section “Pictures of the CCs concentrations”, it should be divided into an independent section.

R3_line 479, “This section may be divided by subheadings. It should provide a concise and precise description of the experimental results, their interpretation as well as the experimental conclusions that can be drawn.” Is this sentence related to the PCA analysis discussion?

The PCA of the data has been eliminated from the revised text.

The results section has been thoroughly revised and is now presented as a single section numbered Section 3.3.

The sentence quoted by the Reviewer was a mistyping error now amended from the revised manuscript.

R3_5.     The quality of Figure 2 is bad, please improve it.

The quality of Figure 2 has been improved. Also a legend to Figure 2 has been added.

Reviewer 4 Report

This study proposed an innovative monitoring approach to measure 14 atmospheric carbonyl compounds (CC) in the vapor and particle phases. Authors showed that this method can avoid the possible artifacts due to the absorption of CC on the particles, increase the collection efficiency, and reduce the possible reactions of target species with the sorbent. Authors also applied principal component analysis (PCA) on the dataset for source attribution. The improved monitoring approach of the CC measurement may benefit the community of pollutant observations. However, several comments related with the description of sampling sites and PCA analysis may need to be addressed before publication.

Fig.2: please specify the representation of roman numbers in the figure at the figure caption.

The abstract mentioned 4 sampling location sites (line 29-30) but the Fig. 2 show more than 4 sampling sites. Please reconcile the difference.

What did the authors expect the difference between Vehicular Emission-dominated samples and roadside samples? They were all mainly influenced by the vehicular emissions according to the sampling protocol.  

PCA is usually used to investigate the sources of the pollutants of interest. It looks that the samples were mainly influenced by vehicular emission based on the experimental design. Since there is only one source for this data set,  what is the main purpose to use PCA in the study? Besides, any reasons for the observation: FA, AA and HEX were most abundant in Viale Pieraccini while BUT, CRO, and iso-VAL were higher in Via Caccini (line 468-470)? Is it because they were influenced by different emission sources in these two locations? And what were the different emission sources that influence Viale Pieraccini and Via Caccini? Which source does the species of AC, BENZ, and VAL represent (Line 473-474)? Are they (AC, BENZ, and VAL) all the chemical marker for vehicular emissions or other sources? Please also specify PCA results in the abstract (line 32-34).

Author Response

Response to reviewer 4  comments

Reviewer 4

This study proposed an innovative monitoring approach to measure 14 atmospheric carbonyl compounds (CC) in the vapor and particle phases. Authors showed that this method can avoid the possible artifacts due to the absorption of CC on the particles, increase the collection efficiency, and reduce the possible reactions of target species with the sorbent. Authors also applied principal component analysis (PCA) on the dataset for source attribution. The improved monitoring approach of the CC measurement may benefit the community of pollutant observations. However, several comments related with the description of sampling sites and PCA analysis may need to be addressed before publication.

The results section has been entirely revised. Additional data have now been presented in order to answer to the reviewers criticisms and a more detailed discussion of the findings has been introduced. The PCA of the data has now been eliminated from the revised text, since we believe that it did not add much to the discussion made by simply looking at the data presented in Tables 1-3.

R4_2) Fig.2: please specify the representation of roman numbers in the figure at the figure caption. Figure 2 has been implemented by adding a detailed legend.

R4_3) The abstract mentioned 4 sampling location sites (line 29-30) but the Fig. 2 show more than 4 sampling sites. Please reconcile the difference.

We studied four sampling locations in Florence, namely the four streets mentioned in the text. The study included more than one sampling site for each street in order to measure as accurately as possible the CCs airborne environmental levels. We thus numbered the replicate sampling sites for each location accordingly in order to let the reader understand that the investigation was comprehensive for each sampling site.

R4_4) What did the authors expect the difference between Vehicular Emission-dominated samples and roadside samples? They were all mainly influenced by the vehicular emissions according to the sampling protocol.

In the revised discussion of the results (Section 3.3) we explained that the difference observed between the two types of sites (VE-dominated and roadside) is due to the traffic transiting the sites, that is different in nature and intensity. The VE-dominated sites are characterized by intense traffic but represented almost totally by cars powered by conventional fuels (LPG, gasoline, diesel), whereas the roadside sites have moderate traffic but due predominantly to heavy vehicles (diesel-fueled).

R4_5) PCA is usually used to investigate the sources of the pollutants of interest. It looks that the samples were mainly influenced by vehicular emission based on the experimental design.

Since there is only one source for this data set, what is the main purpose to use PCA in the study?

We eliminated the PCA of the data of table 1 from the revised text.

R4_6) Besides, any reasons for the observation: FA, AA and HEX were most abundant in Viale Pieraccini while BUT, CRO, and iso-VAL were higher in Via Caccini (line 468-470)?

Is it because they were influenced by different emission sources in these two locations?

And what were the different emission sources that influence Viale Pieraccini and Via Caccini?

Which source does the species of AC, BENZ, and VAL represent (Line 473-474)?

Are they (AC, BENZ, and VAL) all the chemical marker for vehicular emissions or other sources?

We hope that the revised discussion of our results at lines … answered to all the issues raised by the reviewer.

R4_7) Please also specify PCA results in the abstract (line 32-34).

The PCA of the data of table 1 has been eliminated from the revised text.

Round  2

Reviewer 3 Report

The manuscript has been improved a lot.